# GraphHash: Graph Clustering Enables Parameter Efficiency in Recommender Systems

## Abstract

Deep recommender systems rely heavily on large embedding tables to handle high-cardinality categorical features such as user/item identifiers, and face significant memory constraints at scale. To tackle this challenge, hashing techniques are often employed to map multiple entities to the same embedding and thus reduce the size of the embedding tables. Concurrently, graph-based collaborative signals have emerged as powerful tools in recommender systems, yet their potential for optimizing embedding table reduction remains unexplored. This paper introduces `GraphHash`, the first graph-based approach that leverages modularity-based bipartite graph clustering on user-item interaction graphs to reduce embedding table sizes. We demonstrate that the modularity objective has a theoretical connection to message-passing, which provides a foundation for our method. By employing fast clustering algorithms, `GraphHash` serves as a computationally efficient proxy for message-passing during preprocessing and a plug-and-play graph-based alternative to traditional ID hashing. Extensive experiments show that `GraphHash` substantially outperforms diverse hashing baselines on both retrieval and click-through-rate prediction tasks. In particular, `GraphHash` achieves on average a 101.52% improvement in recall when reducing the embedding table size by more than 75%, highlighting the value of graph-based collaborative information for model reduction.

## Keywords

Recommender Systems, Efficiency, Hashing Trick, Graph ML

**ACM Reference Format:**
Anonymous Author(s). 2025. GraphHash: Graph Clustering Enables Parameter Efficiency in Recommender Systems. In *Proceedings of the Web Conference (WWW '25).* ACM, New York, NY, USA, 11 pages. https://doi.org/XXXXXXX.XXXXXXX

## 1 Introduction

The explosive growth of online content has made deep recommender systems (RecSys) essential for content discovery, product suggestions, and targeted advertising in vast digital ecosystems [2, 7, 53]. Large-scale deep RecSys face a critical challenge: their embedding tables, which map each unique value of categorical features like user and item identifiers (IDs) to a dense vector, consume vast amounts of memory due to the high cardinality of these categorical features [13, 19, 22, 28, 33, 34, 51]. For example,

with more than 3B monthly active users worldwide, a single user embedding table for a recommendation model at Meta can easily consume hundreds of GB of memory [22]. This increased memory footprint raises hardware requirements and training costs, potentially limiting model deployability and scalability [13, 19, 22, 51].

To address such a challenge, a common solution is the "*hashing trick*"—assigning IDs to a smaller number of buckets, effectively reducing the embedding table size by having different users or items share the same embedding [13, 19, 33, 41, 46, 51]. A commonly used hashing function in practice is the modulo operation, which assigns entities to buckets solely based on their IDs. While this approach reduces the number of rows in embedding tables, it also introduces *collisions* by forcing dissimilar entities to share the same embedding, leading to significant degradation in model performance [19, 33, 51]. To improve this baseline random hashing, researchers have considered applying the double hashing technique to mitigate collisions, as well as incorporating features and the frequency information [13, 19, 33, 41, 51].

While existing embedding table reduction in RecSys has mostly relied on ID-based or feature-based hashing techniques [13, 19, 28, 33, 41, 51], the field has simultaneously witnessed the rise of collaborative information derived from user-item interactions as a powerful tool. This trend has evolved from classical collaborative filtering to state-of-the-art graph learning approaches [25, 27, 43, 45], highlighting the power of relational data in enhancing recommendation quality. Despite this parallel development, the potential of leveraging collaborative signals for optimizing user/item bucket assignments in model reduction remains largely unexplored. Among various forms of collaborative information, graph representations of user-item interactions have proven particularly effective in recent recommender models. Conventional methods for incorporating this graph information typically rely on *message-passing*, where embeddings are computed by iteratively aggregating and transforming the embeddings of neighboring nodes. This approach effectively captures the rich structural information inherent in user-item interaction graphs, leading to remarkable improvements in recommendation quality. However, message-passing on large-scale graphs involves operations on massive (sparse) matrices, which increase computational requirements and pose challenges for deployment in industry settings where both recommendation quality and efficiency are critical considerations [17, 18, 21, 23, 52, 54].

The above observations motivate the following critical question:

> ***How can we leverage graph information to design hashing functions for embedding table reduction, offering a more efficient alternative to traditional message-passing?***

In this paper, we propose `GraphHash`, a novel approach that leverages modularity-based bipartite graph clustering [5] on the user-item interaction graph to reduce the number of rows in embedding tables. Unlike traditional hash functions, `GraphHash` clusters

the user-item interaction bipartite graph to group "similar" entities based on their interaction patterns, thus generating user/item bucket assignments that better align with the structure of the data when reducing embedding tables. Our choice of modularity-based bipartite graph clustering is motivated by two key factors: first, we demonstrate that based on a random walk interpretation of the modularity maximization objective [5], GraphHash can be regarded as a coarser yet more efficient way to perform the smoothing over the embeddings offered by message-passing, providing a solid foundation for our method. Second, the broad availability of efficient modularity optimization algorithms, such as the Louvain method [6] that employs local greedy heuristics, enables GraphHash to scale effectively to large-scale user-item interaction graphs. As a result, our approach provides a computationally efficient and easily implementable solution for model reduction in large-scale recommender systems by leveraging the user-item interaction graph.

Despite its simplicity in implementation (Algorithm 1), GraphHash achieves substantial performance improvements. It introduces a novel way of utilizing graph information during preprocessing, serving as a scalable and practical alternative to message-passing. This makes GraphHash particularly advantageous for industrial applications where computational and parameter efficiency, combined with ease of implementation, are crucial.

**Our contributions can be summarized as follows:**

- Theoretically, we demonstrate that modularity-based clustering offers a coarser yet more efficient alternative to the smoothing effect of message-passing on embeddings.
- Building on this theoretical insight, we introduce GraphHash, the first graph-based method to effectively utilize the user-item interaction graph for reducing embedding table size. Our approach employs modularity-based bipartite graph clustering, tailored for scalability in large graphs, and acts as a simple, plug-and-play solution for ID hashing in recommender systems. This combination of efficiency and ease of implementation makes GraphHash a practical and powerful tool for improving RecSys performance.
- We conduct extensive evaluations against diverse hashing baselines, showing GraphHash's superior performance in both retrieval and click-through-rate (CTR) prediction tasks[1]. On average, with fewer parameters, GraphHash outperforms the strongest baseline by 101.52% in recall and 88.33% in NDCG for retrieval, while achieving a 2.9% improvement in LogLoss and a 0.2% gain in AUC for CTR.
- Through comprehensive ablation studies across a wide range of experimental settings, we empirically validate our theoretical insights and reveal key findings on the robustness and sensitivity of different design choices in our approach. These results highlight the adaptability and reliability of GraphHash across varying conditions, paving the way for future optimization and refinement in graph-based model reduction.

## 2 Related Work

*Hashing Techniques in RecSys.* Embedding tables, where each row stores the embedding for a user or item, require substantial memory due to the vast number of entities in online platforms. A simple yet effective way to reduce the size of these tables is through the "hashing trick," which randomly hashes unique IDs into a smaller set of values using operations like modulo [46]. Although this approach inevitably leads to collisions, its simplicity has made it widely used in practice. To mitigate collisions, methods such as double hashing and incorporating frequency information have been shown to be important for enhancing model performance [19, 51]. Nonetheless, most prior reduction techniques have focused on ID- or feature-based heuristics, overlooking the user-item interaction information. In this work, we introduce the first graph-based approach for embedding table reduction, integrating interaction information with efficient bipartite graph clustering.

*Graph Clustering.* Graph clustering is a fundamental technique for dimensionality reduction and has been applied in numerous real-world tasks, including more recent ones such as mining higher-order relational data [49], and retrieval-augmented generation in large language models [16]. Common classes of graph clustering include spectral clustering [15, 37], local graph clustering [3] and flow-based clustering [40]. In this work, we choose modularity maximization [5, 32, 36] as the clustering objective, due to its underlying connection with message-passing and computational efficiency, making it well-suited for large-scale graphs in RecSys.

*Graph Learning Beyond Message-Passing.* Graph learning has emerged as a powerful framework for processing relational data [30, 42, 50], with most models following the message-passing paradigm [20], under which node embeddings are computed by recursively aggregating information from all neighboring nodes. However, such a way of integrating the graph in the forward pass also introduces practical challenges, such as scalability with large graphs and oversmoothing, where increasing model depth soon leads to degrading performance [38, 47]. These challenges drive the need for alternative graph learning paradigms. Broadly, existing graph learning methods can be categorized by their use of graphs during preprocessing, training, or inference [54]. In RecSys, traditional collaborative filtering and GNN-based methods use the graph during training [25, 31, 45], while recent methods like TAG-CF leverage it during test-time inference. [27]. Our approach, GraphHash, introduces a novel use of graph structure during preprocessing.

## 3 Method: GraphHash

In this section, we outline the road map leading to our proposed method, GraphHash. To provide context, we begin with a brief overview of representation learning in deep RecSys, where recommendations are generated by interacting user and item embeddings, capturing the essential relationships between entities. This background helps illustrate how bipartite graph structures naturally emerge from user-item interactions in tasks such as retrieval and CTR prediction. We then introduce modularity-based graph clustering, the foundation of our method, which aims to group similar entities based on interaction patterns. Finally, we explore the connection between the modularity maximization objective and message-passing techniques, offering deeper theoretical insights into the mechanics underlying GraphHash.

### 3.1 Embedding in Deep RecSys

Recommendations are generated by "interacting" user and item embeddings, typically through computing the dot product between

---

[1]The implementation of GraphHash along with all baselines and backbones are available at https://anonymous.4open.science/r/GraphHash2024-1BD8.

corresponding rows of the user and item embedding matrices [19]. In deep RecSys, these embeddings are learned representations that map high-dimensional data—such as user preferences or item characteristics—into lower-dimensional vectors, capturing the essential relationships between users and items. These representations are stored in embedding tables, with separate tables for users and items.

The embeddings play a central role in two key tasks: retrieval and click-through rate (CTR) prediction. For retrieval, the system suggests relevant items by comparing the similarity between user and item embeddings, while CTR prediction estimates the likelihood of user engagement with a specific item. Both tasks rely heavily on modeling user-item interactions, often represented as a bipartite graph, where users and items form the nodes, and interactions such as clicks, purchases, or ratings form the edges [25, 43, 45].

With the huge number of entities in online platforms, embedding tables modern recommender systems can easily take hundreds of GB of memory footprint. While commonly adopted hashing tricks [19, 51] can effectively reduce the number of rows, the undesired collisions would negatively affect the recommendation accuracy. Therefore, aside from mitigating collisions by mapping entities to a larger set [51], another effective solution would be to map "similar" entities—those with similar interaction patterns—into the same embedding [19]. Graph clustering, which effectively groups entities based on their interaction patterns, can help achieve this, reducing the impact of collisions while preserving recommendation quality.

## 3.2 Modularity-based Graph Clustering

To implement our clustering approach, we rely on modularity, a widely used objective for graph clustering [5, 6, 14, 32, 36, 40]. Modularity-based clustering groups similar entities based on the density of their connections, ensuring that densely connected entities share the same embedding. Specifically, for clustering the user-item bipartite graph, we adopt the modularity definition for bipartite graphs proposed in [5]: given the set of users $\mathcal{U} \subset \mathbb{N}$ and set of items $\mathcal{I} \subset \mathbb{N}$, the adjacency matrix $A \in \mathbb{R}^{|\mathcal{U}| \times |\mathcal{I}|}$ of the user-item bipartite graph $\mathcal{G}(\mathcal{U}, \mathcal{I}, \mathcal{E})$, which encodes the set of user-item interaction pairs $\mathcal{E}$, is defined as

$$A_{ui} = \begin{cases} 1 & (u, i) \in \mathcal{E} \\ 0 & otherwise. \end{cases}$$

Then *modularity* of a cluster assignements $\mathcal{P}$ for the bipartite graph $\mathcal{G}$ is defined as

$$Q = \frac{1}{m} \sum_{C \in \mathcal{P}} \sum_{u,i \in C} \left( A_{ui} - \frac{k_u d_i}{m} \right),$$

where $m = |\mathcal{E}|$ is the number of edges in $\mathcal{G}$, $k_u = \sum_j A_{uj}$ is the degree of user $u$, and $d_i = \sum_j A_{ji}$ is the degree of item $i$. The optimal cluster assignments $\mathcal{P}^*$ in terms of modularity is then found by maximizing $Q$.

Directly optimizing modularity is NP-hard [9]. In practice, optimal partitions can be found by modularity optimization algorithms. One of the most popular and state-of-the-art modularity optimization method is the Louvain method [6], which is based on greedy heuristics and enables efficient clustering even on graphs with billions of nodes [6].[2] Denote the algorithm as $\mathcal{A}$, then the clustering assignment of node $x$ is given by $\mathcal{A}(x)$.

---

**Algorithm 1** Example GraphHash implementation with Louvain

```
""" ID hashing """
louvain = Louvain(resolution=resolution)
louvain.fit(train_data, force_bipartite=True)
user_hashed_id = map_to_consec_int(louvain.labels_row_)
item_hashed_id = map_to_consec_int(louvain.labels_col_)
""" build model with hashed user/item ID vocab """
user_vocab = np.unique(user_clusters)
item_vocab = np.unique(user_clusters)
model = MFRetriever(user_vocab, item_vocab, emb_dim)
""" model training """
for user_id, item_id in train_data:
    pos_score = model(user_hashed_id[user_id],
                      item_hashed_id[item_id])
    ... # negative sampling, BPR loss, etc
```

---

## 3.3 GraphHash

With the clustering assignments obtained through modularity-based graph clustering, we can now extend this approach to define the hashing mechanism of GraphHash. Given the set of users $\mathcal{U} \subset \mathbb{N}$ and items $\mathcal{I} \subset \mathbb{N}$, a hash function $\mathcal{H}$ assigns these IDs to a smaller set of buckets, $\mathcal{B} \subset \mathbb{N}$. In this setup, users or items within the same bucket will share the same embedding in the corresponding embedding table.

The clustering assignments provided by the modularity optimization algorithm $\mathcal{A}$ offer a natural way to define these bucket assignments. By leveraging the dense connections between users and items—reflecting similar behaviors or preferences—we can improve recommendation quality while maintaining the memory budget. Formally, the bucket assignments are derived from the cluster assignments $\mathcal{A}(\mathcal{U})$ and $\mathcal{A}(\mathcal{I})$. To ensure consistent and ordered assignments, a relabeling function $\ell$ maps the clusters to consecutive integers based on the order of their appearance in $\mathcal{A}(\mathcal{U})$ and $\mathcal{A}(\mathcal{I})$. GraphHash can then be defined as:

$$\text{GraphHash}(x) = \ell(\mathcal{A}(x)), \quad \forall x \in \mathcal{U}, \mathcal{I}. \tag{1}$$

Algorithm 1 gives an example pseudocode for implementing GraphHash with the Louvain algorithm on a matrix factorization retriever. This approach requires minimal changes to existing code and can be easily integrated in a plug-and-play manner into any recommender model that uses embedding tables.

While graph clustering differs from traditional hash functions in various ways, one key property of regular hash functions that GraphHash shares is that given the user-item interaction graph, it is deterministic when $\mathcal{A}$ is the Louvain algorithm:

PROPOSITION 3.1. *Given $\mathcal{G}(\mathcal{U}, \mathcal{I}, \mathcal{E})$, where $\mathcal{U}, \mathcal{I}$ are finite subsets of $\mathbb{N}$, and $\mathcal{A}$ is the Louvain algorithm. Then* $\text{GraphHash}(\cdot)$ : $\mathcal{U}, \mathcal{I} \to \{1, 2, ..., |\mathcal{P}^*|\}$ *is a deterministic function.*

---

[2]In the implementation of our method, we make use of the function provided in the scikit-network library [8].

The proof can be found in Appendix A. As such, one advantage of `GraphHash` is that it behaves like a regular hash function, making it easy to integrate with existing techniques such as double hashing, which was proposed to reduce the collision rate between embeddings of different entities during hashing [51]. By using a regular random hash function $\mathcal{H}$ and `GraphHash`, we derive a natural variant of our method to improve collision mitigation, which we referred as `DoubleGraphHash`:

$$\text{DoubleGraphHash}(x) = (\mathcal{H}(x), \text{GraphHash}(x)), \forall x \in \mathcal{U}, \mathcal{I}. \quad (2)$$

Similarly as discussed in [51], the combination of $\mathcal{H}$ and `GraphHash` can be viewed as an approximation of a hashing into a set of larger cardinality to mitigate collisions between embeddings of different entities when reducing the number of rows in a embedding table.

### 3.4 Why Modularity? A Random Walk Perspective

While various graph clustering methods exist, the use of modularity-based graph clustering as an alternative to hashing has a fundamental, albeit implicit, connection with message-passing techniques that have proven effective in RecSys models [25, 45]. The link becomes apparent when considering the random walk interpretation of modularity [14, 32]: under modularity, an optimal clustering assignment is one where a random walker is the most likely to remain within its starting cluster compared to chance. Based on this criterion, modularity can be rewritten in the following expressions:

$$Q = \sum_{C \in \mathcal{P}} \sum_{u,i \in C} \left( \frac{A_{ui}}{d_i} \frac{d_i}{m} - \frac{k_u d_i}{m^2} \right) = \sum_{C \in \mathcal{P}} \sum_{u,i \in C} \left( \frac{A_{iu}}{k_u} \frac{k_u}{m} - \frac{k_u d_i}{m^2} \right).$$

Essentially, modularity $Q$ computes the probability of starting in a cluster $C$, and still being in a cluster $C$ after one step of unbiased random walk minus the probability that two independent random walkers are in $C$, evaluated at large-time asymptotic.

On the other hand, one iteration of message-passing can be written as

$$X'_{\mathcal{U}} = D_{\mathcal{U}}^{-1/2} A D_{\mathcal{I}}^{-1/2} X_{\mathcal{I}}, \qquad X'_{\mathcal{I}} = D_{\mathcal{I}}^{-1/2} A^{\top} D_{\mathcal{U}}^{-1/2} X_{\mathcal{U}}, \quad (3)$$

where $X_{\mathcal{U}}, X_{\mathcal{I}}$ are the user and item embeddings, respectively, and $D_{\mathcal{U}}, D_{\mathcal{I}}$ are the diagonal degree matrices for users and items, respectively. This operation essentially recursively smoothes a node's embedding with the embeddings of its neighboring nodes [25, 48]. From a random walk perspective, one can directly interpret message-passing in (3) as a random walker starting from each root node, then update the root node's embedding with the embeddings of other nodes in the reachable neighborhood, weighted by the corresponding unbiased random walk transition probabilities $D^{-1}A$ (up to a left and right matrix transformation at both ends): $X' = D^{1/2}(D^{-1}A)D^{-1/2}X$. However, a natural question to pose for the message-passing process is: when should the random walker stop, i.e., which neighbors should each node use for smoothing? The number of message-passing layers in a model directly affects the smoothness of the embeddings, which in turn impacts downstream task performance. Yet message-passing methods such as LightGCN treat it as a hyperparameter requiring manual tuning on a case-by-case basis to achieve optimal results [25].

Under this random walk interpretation of modularity, `GraphHash` can be seen as a coarser but more efficient way to perform smoothing over the graph, similar to iterative message-passing. There are two key differences: 1) rather than being set as a hyperparameter, the random walk's stopping point is now automatically determined by maximizing modularity so that the probability of staying in the starting cluster is maximized; 2) `GraphHash` fully smooths node embeddings within the same cluster, while message-passing gradually smooths embeddings through the iterative process in (3). Although this approach sacrifices some granularity in node embeddings compared to iterative message-passing, this trade-off allows for greater computational efficiency: `GraphHash` simplifies the process by fully smoothing node embeddings within the same cluster in a single step, rather than iteratively computing them over multiple layers.

## 4 Research Questions

We are interested in investigating the following aspects of `GraphHash`:

**RQ1** How does hashing based on the graph information compared with pure ID-based or feature-based hashing methods?

**RQ2** Is the graph information more beneficial to power or tail users?

**RQ3** How would the training objective affect the model performance with hashing?

**RQ4** How would hashing based on the graph information help if the backbone model also uses the graph information?

**RQ5** How would different graph clustering objectives affect the model performance?

## 5 Evaluation of GraphHash's Effectiveness

In this section, we validate the effectiveness of our proposed `GraphHash`, and answer **RQ1** and **RQ2** above.

### 5.1 Experimental Setup

We benchmark all hashing methods for embedding table reduction on two key recommendation tasks: context-free top-k retrieval and context-aware click-through-rate (CTR) prediction. Here, context-free means that models do not use any additional feature information other than the IDs of users or items, whereas context-aware models utilize complimentary contextual features in addition to the user or item IDs [41]. Due to the nature of our method, we select publicly available datasets where user ID and item ID are explicitly available. Namely, Gowalla [12], Yelp2018 and AmazonBook [1] for retrieval, and Frappe [4], MovieLens-1M, and MovieLens-20M [24] for CTR. Further details on datasets are provided in Appendix B.

*5.1.1 Backbones.* We use matrix factorization (MF) [31], Neural Matrix Factorization (NeuMF) [26], LightGCN [25], and MF+DirectAU (DAU) loss [43] as backbones for the retrieval task, where the first three are trained with the Bayesian Personalized Ranking (BPR) loss [39]; we use WideDeep [10], DLRM [35], and DCNv2 [44], all trained with binary cross entropy loss (LogLoss) for the CTR task.

*5.1.2 Baselines.* We evaluate `GraphHash` against the following baseline hashing methods:

- **Random:** we apply modulo operation to IDs.
- **Frequency** [19, 51]: we allocate half the number of buckets to individual users/items with the highest frequencies in the training data, and apply random hashing to the rest.

- **Double** [51]: we apply two hash functions to IDs and generate two hash codes for each entity and sum the corresponding entries in the embedding table up as the embedding for the entity.
- **Double frequency** [51]: similar to frequency, we allocate half the number of buckets to individual users/items that have the highest frequencies in the training data. We then apply double hashing to the rest of the entities.
- **LSH**: we apply locally sensitive hashing (LSH) to user/item features, if features are available.
- **LSH-structure**: we treat one-hop neighbor patterns in the user-item interaction graph as the features ($\mathcal{A}$ as the feature matrix for users and $\mathcal{A}^\top$ as the feature matrix for items) and apply LSH hashing. This can been seen as an alternative way to use the graph structure for user/item bucket assignments.

For reference, we also include the results of models without hashing (**full**).

We implemented all the backbones, baselines, and our approaches with PyTorch. For a fair comparison, all the implementations were identical across all the models except for the hashing component and the resulting embedding table. More details about the experimental setup and training can be found in Appendix C.

## 5.2 Performance Comparison (RQ1)

*5.2.1 Performance in retrieval task.* Table 1 reports the mean and standard deviation of the standard retrieval evaluation metrics, Recall@20 and NDCG@20 (in percentage), over 5 independent runs using the best hyperparameters. We see that our proposed method GraphHash achieves the best performance across datasets and backbones, and the improvements over the strongest baseline are substantial, with an average of 101.52% increase in Recall@20 and 88.33% in NDCG@20.

The only exception occurs in the Yelp2018 dataset when employing MF+DirectAU loss as the backbone, where our method slightly underperforms compared to double frequency hashing. Notably, all hashing methods exhibit a significant performance drop when transitioning from BPR loss to DirectAU loss. We conduct a thorough examination of this phenomenon in Section 6.1, where we present a detailed ablation study on the DirectAU loss function and its impact on the model performance.

*5.2.2 Performance in CTR task.* Table 2 reports the mean and standard deviation of the standard CTR evaluation metrics, LogLoss and AUC (Area Under the ROC Curve), over 5 independent runs using the best hyperparameters. Unlike the case for the retrieval task, our proposed method GraphHash does not perform as ideal. We then further consider a variant of our method, DoubleGraphHash in (2), which combines GraphHash with another random hashing function based on the double hashing technique [51] to mitigate collisions. We see that DoubleGraphHash achieves much better performance than GraphHash on CTR tasks and in fact, the best performance across datasets and backbones.

*5.2.3 The impact of collisions on retrieval vs. CTR performance.* Comparing the results for retrieval and CTR tasks, we make the following observations: 1) GraphHash performs the best in the retrieval task but falls short in the CTR task; 2) DoubleGraphHash, which incorporates double hashing, is the top performer for the CTR task; and 3) while double hashing methods underperform in the retrieval task, they are much stronger baselines for CTR. These findings suggest that pure user-item interaction information is more directly beneficial for retrieval, where collision is less of an issue. In contrast, for the CTR task, collision avoidance techniques are essential for improving performance.

## 5.3 User Subgroup Evaluation (RQ2)

Next, we examine model performance across different user groups, categorized by their frequency percentile in the training data. For the retrieval task, we aggregate the metrics within each degree subgroup. For the CTR task, we divide the clicks in the test set based on the user subgroup that generated the click. Figures 1 and 3 showcase the results for the retrieval and CTR tasks, respectively.

For the retrieval task, incorporating frequency information generally benefits power users, regardless of the backbone model used. In contrast, GraphHash achieves more balanced performance across all user groups, closely resembling the trend of models without hashing. For the CTR task, all methods, including those without hashing, tend to perform better for clicks generated by power users. Notably, DoubleGraphHash, which delivers the best overall performance, also performs better for power users than for tail users. These observations suggest the fundamental differences between the retrieval and CTR tasks. Nevertheless, the user-item interaction graph benefits model performance in both tasks, with different variants of the method optimizing for their specific characteristics.

## 6 Method Analysis

In this section, we conduct further ablation studies investigating various aspects of our approach, providing deeper insights into its function, robustness and adaptability across different scenarios.

## 6.1 The Impact of Training Objective (RQ3)

For the retrieval task, we have considered two popular loss functions when training the backbone MF model: the BPR loss and the DirectAU loss. As shown in Table 1, while DirectAU performs better on MF models without hashing, the performance drops significantly for all hashing-based methods when switching from BPR to DirectAU. This finding aligns with recent results in the literature, suggesting that DirectAU may not be compatible with hashing-based methods [41].

To further investigate this phenomenon, we conduct an additional set of experiments with varying values of $\gamma$ in $[0.25, 0.5, 1, 2, 5]$[3], the strength of the uniformity term in the DirectAU loss, and compare the model performance without hashing, with double frequency hashing (the strongest baseline), and with GraphHash in terms of Recall@20 on Gowalla and Yelp2018. The results in Figure 2 show that while both the model without hashing and GraphHash are quite robust to changes in $\gamma$, there exists a specific sweet spot for the value of $\gamma$ under double frequency hashing. The corresponding results in NDCG@20 can be found in Appendix D and exhibit similar trends. This indicates that although hashing methods may generally be less compatible with DirectAU than with BPR (Table 1), GraphHash, by leveraging graph information, is more robust to the choice of $\gamma$.

---

[3]The same range considered in the original DirectAU paper [43].

**Table 1: Benchmark performance on the top-20 item retrieval task (values in percentage). The best performance is highlighted in bold, with the second best underlined. Our method substantially outperforms the strongest baseline, achieving on average a 101.52% improvement in Recall@20 and an 88.33% improvement in NDCG@20. Note that we deliberately control the number of rows in the embedding tables of GraphHash to be smaller than all the baselines, while keeping the rest of backbone models identical, to emphasize the point that GraphHash obtains improvements even with shorter embedding tables.**

|  |  | Gowalla | | | Yelp2018 | | | AmazonBook | | |
|---|---|---|---|---|---|---|---|---|---|---|
|  |  | # params | Recall@20 (↑) | NDCG@20 (↑) | # params | Recall@20 (↑) | NDCG@20 (↑) | # params | Recall@20 (↑) | NDCG@20 (↑) |
| MF | full | 4.534M | 15.617±0.133 | 9.661±0.096 | 4.462M | 7.779±0.069 | 4.951±0.036 | 9.231M | 7.711±0.197 | 4.951±0.111 |
|  | random | 0.744M | 0.766±0.019 | 0.382±0.017 | 0.977M | 0.546±0.012 | 0.315±0.012 | 1.258M | 0.185±0.007 | 0.113±0.005 |
|  | double | 0.744M | 2.618±0.086 | 1.604±0.073 | 0.977M | 1.355±0.111 | 0.850±0.034 | 1.258M | 0.767±0.027 | 0.525±0.017 |
|  | frequency | 0.744M | 3.679±0.043 | 2.429±0.016 | 0.977M | 2.401±0.034 | 1.624±0.020 | 1.258M | 1.287±0.020 | 0.918±0.013 |
|  | double frequency | 0.744M | 3.888±0.055 | 2.532±0.041 | 0.977M | 2.478±0.032 | 1.665±0.027 | 1.258M | 1.311±0.023 | 0.919±0.021 |
|  | LSH-structure | 1.049M | 1.106±0.054 | 0.617±0.035 | 1.049M | 0.729±0.038 | 0.460±0.018 | 2.097M | 0.452±0.033 | 0.290±0.024 |
|  | GraphHash (ours) | 0.742M | **9.300**±0.064 | **5.512**±0.019 | 0.976M | **3.699**±0.077 | **2.325**±0.040 | 1.255M | **3.970**±0.045 | **2.667**±0.037 |
| NeuMF | full | 9.084M | 13.305±0.104 | 8.096±0.076 | 8.940M | 6.260±0.088 | 3.805±0.062 | 18.480M | 6.452±0.163 | 4.098±0.109 |
|  | random | 1.505M | 0.926±0.022 | 0.437±0.011 | 1.971M | 0.594±0.037 | 0.334±0.023 | 2.533M | 0.203±0.019 | 0.131±0.013 |
|  | double | 1.505M | 4.718±0.239 | 2.934±0.169 | 1.971M | 2.945±0.082 | 1.777±0.060 | 2.533M | 1.316±0.036 | 0.815±0.036 |
|  | frequency | 1.505M | 3.518±0.027 | 2.256±0.027 | 1.971M | 2.269±0.051 | 1.472±0.037 | 2.533M | 1.187±0.029 | 0.821±0.062 |
|  | double frequency | 1.505M | 4.477±0.079 | 2.719±0.140 | 1.971M | 2.941±0.176 | 1.753±0.101 | 2.533M | 1.415±0.031 | 0.926±0.028 |
|  | LSH-structure | 2.114M | 1.098±0.121 | 0.608±0.093 | 2.114M | 0.648±0.127 | 0.399±0.097 | 4.211M | 0.445±0.032 | 0.289±0.020 |
|  | GraphHash (ours) | 1.501M | **7.886**±0.084 | **4.565**±0.049 | 1.970M | **3.150**±0.084 | **1.969**±0.051 | 2.527M | **3.842**±0.049 | **2.477**±0.028 |
| LightGCN | full | 4.534M | 18.321±0.348 | 11.534±0.096 | 4.462M | 8.889±0.079 | 5.549±0.048 | 9.231M | 8.471±0.342 | 5.501±0.236 |
|  | random | 0.744M | 9.214±0.033 | 5.909±0.022 | 0.977M | 4.894±0.017 | 3.114±0.010 | 1.258M | 2.485±0.074 | 1.704±0.047 |
|  | double | 0.744M | 9.277±0.140 | 6.012±0.071 | 0.977M | 5.305±0.103 | 3.359±0.080 | 1.258M | 2.410±0.039 | 1.662±0.022 |
|  | frequency | 0.744M | 8.574±0.017 | 5.480±0.019 | 0.977M | 4.931±0.031 | 3.142±0.017 | 1.258M | 2.184±0.021 | 1.497±0.013 |
|  | double frequency | 0.744M | 9.880±0.199 | 6.414±0.135 | 0.977M | 5.987±0.030 | 3.855±0.030 | 1.258M | 2.783±0.027 | 1.869±0.018 |
|  | LSH-structure | 1.049M | 9.780±0.041 | 6.291±0.060 | 1.049M | 4.789±0.034 | 3.025±0.031 | 2.097M | 3.279±0.048 | 2.199±0.020 |
|  | GraphHash (ours) | 0.742M | **15.325**±0.127 | **9.658**±0.054 | 0.976M | **6.244**±0.088 | **3.882**±0.054 | 1.255M | **7.261**±0.034 | **5.033**±0.023 |
| MF + DAU | full | 4.534M | 17.984±0.055 | 11.633±0.020 | 4.462M | 11.081±0.021 | 7.207±0.013 | 9.231M | 10.264±0.025 | 6.899±0.015 |
|  | random | 0.744M | 0.478±0.031 | 0.260±0.016 | 0.977M | 0.429±0.008 | 0.255±0.007 | 1.258M | 0.159±0.003 | 0.100±0.003 |
|  | double | 0.744M | 0.551±0.051 | 0.362±0.032 | 0.977M | 0.401±0.017 | 0.263±0.014 | 1.258M | 0.188±0.010 | 0.124±0.008 |
|  | frequency | 0.744M | 1.487±0.007 | 1.305±0.015 | 0.977M | 1.107±0.019 | 0.980±0.021 | 1.258M | 0.797±0.013 | 0.758±0.016 |
|  | double frequency | 0.744M | 3.440±0.121 | 2.677±0.074 | 0.977M | **2.536**±0.033 | **1.903**±0.025 | 1.258M | 1.466±0.026 | 1.218±0.028 |
|  | LSH-struc | 1.049M | 0.809±0.007 | 0.433±0.013 | 1.049M | 0.419±0.025 | 0.252±0.010 | 2.097M | 0.390±0.018 | 0.236±0.009 |
|  | GraphHash (ours) | 0.742M | **7.660**±0.136 | **4.148**±0.096 | 0.976M | 2.458±0.049 | 1.414±0.033 | 1.255M | **4.602**±0.012 | **3.107**±0.010 |

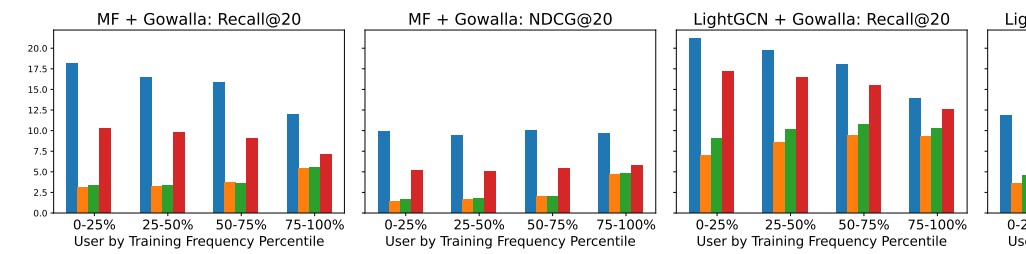

**Figure 1: Performance breakdown of the retrieval task by test user frequency in the training data. Frequency information tends to benefit power users, regardless of the backbone model. In contrast, GraphHash achieves balanced performance across all user groups, closely mirroring the trend of the full model.**

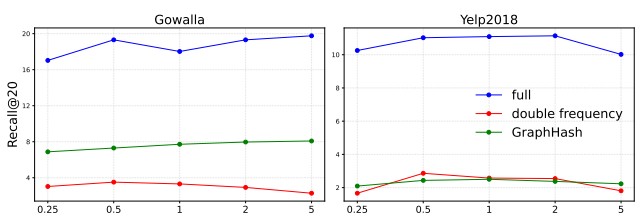

**Figure 2: Impact of the uniformity term $\gamma$ in DirectAU on model performance. While the full model and GraphHash are robust to changes in $\gamma$, double frequency hashing shows a sweet spot, suggesting GraphHash enhances robustness to $\gamma$ in hashing methods.**

## 6.2 The Impact of Backbone GNN's Depth (RQ4)

Given the theoretical connection between modularity-based graph clustering and message-passing discussed in Section 3.4, we empirically study the impact of the depth of the backbone GNN, specifically LightGCN here, on the performance of GraphHash. We vary the depth of the LightGCN backbone model and the resulting performance without hashing, with random hashing, and with GraphHash in terms of Recall@20 on Gowalla and Yelp2018 are presented in Figure 4. The corresponding results in NDCG@20 can be found in Appendix D, which show similar trends. We see that while all these three methods' performance improve with the increasing backbone depth, it tends to saturate after 3-4 layers, aligning with the observation in the original LightGCN paper [25]. Yet GraphHash consistently outperforms random hashing at all depth levels and is able

Table 2: Benchmark performance on the CTR task. The best performance is highlighted in bold, with the second best underlined. On average, **DoubleGraphHash** reduces LogLoss by 0.008 and improves AUC by 0.002. Note that in high-precision tasks such as CTR prediction, even small improvements can lead to substantial performance enhancements and significant business gains at scale [10, 44].

| | | Frappe | | | MovieLens-1M | | | MovieLens-20M | | |
|---|---|---|---|---|---|---|---|---|---|---|
| | | # params | LogLoss (↓) | AUC (↑) | # params | LogLoss (↓) | AUC (↑) | # params | LogLoss (↓) | AUC (↑) |
| WideDeep | full | 0.509M | 0.248±0.012 | 0.980±0.001 | 0.695M | 0.317±0.002 | 0.896±0.001 | 5.267M | 0.321±0.000 | 0.895 ±0.000 |
| | random | 0.221M | 0.352±0.008 | 0.928±0.000 | 0.118M | 0.372±0.002 | 0.850±0.000 | 0.744M | 0.342±0.001 | 0.878±0.000 |
| | double | 0.221M | 0.478±0.140 | 0.968±0.001 | 0.118M | 0.359±0.000 | 0.860±0.002 | 0.744M | 0.340±0.001 | 0.880±0.000 |
| | frequency | 0.221M | 0.283±0.005 | 0.956±0.001 | 0.118M | 0.382±0.002 | 0.842±0.002 | 0.744M | 0.342±0.001 | 0.878±0.001 |
| | double frequency | 0.221M | 0.356±0.055 | 0.970±0.001 | 0.118M | 0.365±0.003 | 0.855±0.002 | 0.744M | 0.339±0.000 | 0.880±0.000 |
| | LSH | - | - | - | 0.141M | 0.481±0.000 | 0.704±0.001 | 0.793M | 0.446±0.000 | 0.767±0.000 |
| | LSH-structure | 0.252M | 0.342±0.028 | 0.932±0.008 | 0.141M | 0.376±0.002 | 0.849±0.001 | 1.094M | 0.349±0.002 | 0.872±0.002 |
| | GraphHash (ours) | 0.218M | 0.286±0.010 | 0.949±0.001 | 0.115M | 0.384±0.001 | 0.841±0.001 | 0.744M | 0.345±0.001 | 0.875±0.000 |
| | DoubleGraphHash (ours) | 0.218M | **0.265**±0.060 | **0.972**±0.001 | 0.115M | **0.358**±0.001 | **0.861**±0.001 | 0.744M | **0.337**±0.001 | **0.881**±0.001 |
| DLRM | full | 0.443M | 0.194±0.022 | 0.982±0.001 | 0.725M | 0.322±0.002 | 0.894±0.001 | 5.258M | 0.321±0.001 | 0.893±0.001 |
| | random | 0.155M | 0.351±0.009 | 0.929±0.002 | 0.148M | 0.373±0.001 | 0.848±0.001 | 0.735M | 0.345±0.001 | 0.876±0.001 |
| | double | 0.155M | 0.246±0.029 | 0.970±0.001 | 0.148M | 0.363±0.002 | 0.858±0.001 | 0.735M | 0.341±0.001 | 0.879±0.001 |
| | frequency | 0.155M | 0.270±0.011 | 0.956±0.000 | 0.148M | 0.381±0.001 | 0.841±0.001 | 0.735M | 0.341±0.000 | 0.878±0.000 |
| | double frequency | 0.155M | 0.291±0.021 | 0.969±0.001 | 0.148M | 0.375±0.006 | 0.848±0.004 | 0.735M | 0.343±0.001 | 0.876±0.001 |
| | LSH | - | - | - | 0.171M | 0.481±0.001 | 0.704±0.001 | 0.784M | 0.448±0.001 | 0.764±0.001 |
| | LSH-structure | 0.186M | 0.358±0.022 | 0.928±0.007 | 0.171M | 0.384±0.007 | 0.840±0.005 | 1.085M | 0.352±0.001 | 0.869±0.001 |
| | GraphHash (ours) | 0.152M | 0.278±0.003 | 0.950±0.001 | 0.145M | 0.383±0.002 | 0.840±0.001 | 0.735M | 0.347±0.001 | 0.873±0.001 |
| | DoubleGraphHash (ours) | 0.152M | **0.231**±0.011 | **0.973**±0.001 | 0.145M | **0.361**±0.002 | **0.860**±0.001 | 0.735M | **0.339**±0.000 | **0.880**±0.000 |
| DCNv2 | full | 1.289M | 0.144±0.017 | 0.981±0.002 | 0.746M | 0.310±0.001 | 0.900±0.001 | 5.290M | 0.322±0.001 | 0.894±0.001 |
| | random | 1.001M | 0.320±0.004 | 0.930±0.001 | 0.169M | 0.366±0.001 | 0.855±0.001 | 0.767M | 0.338±0.000 | 0.881±0.000 |
| | double | 1.001M | 0.217±0.008 | 0.968±0.001 | 0.169M | 0.358±0.000 | 0.862±0.001 | 0.767M | 0.338±0.000 | 0.882±0.001 |
| | frequency | 1.001M | 0.241±0.003 | 0.956±0.000 | 0.169M | 0.375±0.000 | 0.846±0.001 | 0.767M | 0.337±0.000 | 0.881±0.000 |
| | double frequency | 1.001M | 0.223±0.016 | 0.968±0.002 | 0.169M | 0.361±0.001 | 0.860±0.001 | 0.767M | 0.337±0.000 | 0.881±0.000 |
| | LSH | - | - | - | 0.190M | 0.480±0.000 | 0.704±0.000 | 0.817M | 0.443±0.000 | 0.772±0.000 |
| | LSH-structure | 1.032M | 0.367±0.016 | 0.928±0.002 | 0.192M | 0.368±0.003 | 0.852±0.003 | 1.117M | 0.346±0.001 | 0.875±0.001 |
| | GraphHash (ours) | 0.998M | 0.263±0.001 | 0.951±0.001 | 0.166M | 0.380±0.001 | 0.843±0.000 | 0.767M | 0.343±0.001 | 0.877±0.001 |
| | DoubleGraphHash (ours) | 0.998M | **0.194**±0.003 | **0.972**±0.001 | 0.166M | **0.356**±0.001 | **0.864**±0.000 | 0.767M | **0.337**±0.000 | **0.882**±0.001 |

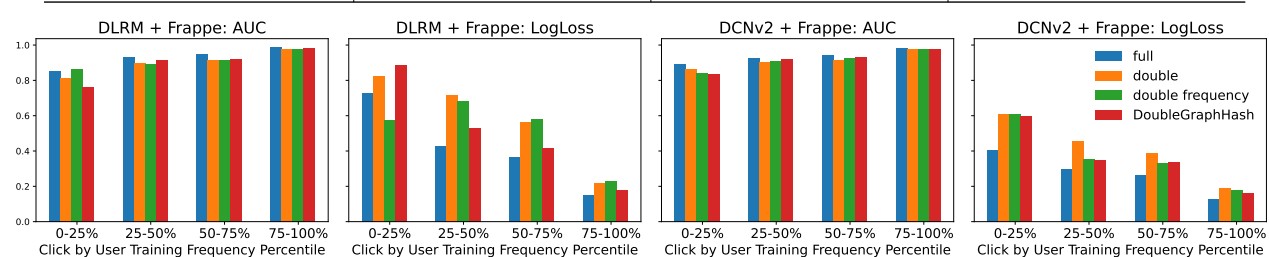

Figure 3: Performance breakdown of the CTR task by user frequency in training data. All methods tend to perform better for clicks generated by power users, and **DoubleGraphHash**, which obtains the best overall performance, also works better for clicks generated by power users than for those generated by tail users.

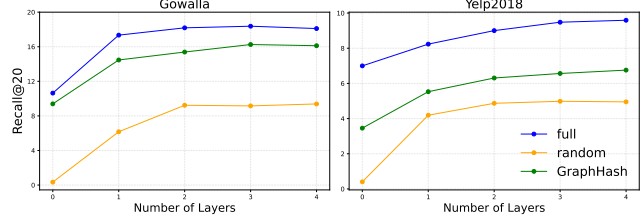

Figure 4: The impact of LightGCN's depth on the performance of different hashing methods. **GraphHash** consistently outperforms random hashing. In particular, **GraphHash** without any additional message-passing layers, performs roughly equal to random hashing with one or two message-passing layers, where the performance in the latter model can be sorely attributed to pure message-passing.

to recover roughly a similar percentage of the model performance without hashing at each depth. Most interestingly, **GraphHash** without any additional message-passing layers, performs roughly equal to random hashing with 1 or 2 message-passing layers, where the

performance in the latter model can be sorely attributed to message-passing. This is in line with our theory that **GraphHash** can be seen as a coarser but more efficient way to achieve a similar smoothing effect to message-passing, at the alternative cost of performing graph clustering during preprocessing.

## 6.3 Analysis of Embedding Smoothness

The smoothing effect of message-passing has been identified to be helpful for node level tasks [48] and particularly for the effectiveness of LightGCN in retrieval [25]. From the smoothing perspective, there are two key differences between **GraphHash** and message-passing: 1) in GNNs such as LightGCN, the number of message-passing layers is a hyperparameter that requires manual tuning [25, 45], whereas the modularity maximization objective in **GraphHash** automatically find the best neighborhood for each node to perform smoothing. 2) In GNNs, smoothing is done through iteratively applying message-passing, whereas in **GraphHash**, once the optimal neighborhood is found for each node, embeddings are smoothed to be identical within each neighborhood.

**Table 3: Average within cluster smoothness found by learning in full models without hashing. Compared to the 2-hop neighborhood for each node, GraphHash finds a better candidate neighborhood to perform complete smoothing.**

| | | Gowalla | | Yelp2018 | |
|---|---|---|---|---|---|
| | | $\mathcal{S}(X_{\mathcal{U}},C)$ | $\mathcal{S}(X_{\mathcal{I}},C)$ | $\mathcal{S}(X_{\mathcal{U}},C)$ | $\mathcal{S}(X_{\mathcal{I}},C)$ |
| MF | 2-hop | 15.046 | 12.345 | 3.987 | 3.718 |
| | GraphHash | 8.324 | 7.628 | 1.848 | 1.592 |
| LightGCN | 2-hop | 70.653 | 46.458 | 86.977 | 68.900 |
| | GraphHash | 35.462 | 28.080 | 49.712 | 42.2439 |

We further investigate whether the user/item clusters found by GraphHash, in which the embeddings of different users/items would be fully smoothed, corresponds to good candidates found by learning when the model has enough capacity, i.e. without hashing. Here, we measure the average within cluster smoothness, normalized by the number of entities in each cluster, for a cluster assignment $C$ on user embeddings $X_{\mathcal{U}}$ by

$$\mathcal{S}(X_{\mathcal{U}},C) = \frac{1}{|\mathcal{U}|} \sum_{u \in \mathcal{U}} \sum_{u' \in C(u)} \frac{\|X_u - X_{u'}\|_2^2}{|C(u)|},$$

and similarly for item embeddings $X_{\mathcal{I}}$ by $\mathcal{S}(X_{\mathcal{I}},C)$.

We compare the cluster assignments $C$ given by GraphHash, against the two-hop neighborhood for each node, which corresponds to the neighborhoods for smoothing when applying two message-passing layers. The results computed on the embeddings of MF and LightGCN without hashing on Gowalla and Yelp2018 are shown in Table 3. We make the following two observations: 1) GraphHash indeed finds a better candidate neighborhood to perform complete smoothing, as compared to the 2-hop neighborhood for each node. This also makes sense as message-passing would not completely smooth the embeddings within the 2-hop neighborhood for each node. 2) Compared to the ones in MF, the embeddings in LightGCN are less smooth, since message-passing would perform further smoothing over them.

## 6.4 The Impact of Clustering Objective (RQ5)

As discussed in Section 3.4, the choice of the modularity objective for clustering is based on its theoretical connection to message-passing and its computational efficiency in practice. In this section, we study how the clustering objective affects the model performance in terms of accuracy and efficiency.

*6.4.1 Modularity-based clustering at varying resolution.* In Section 3.4, we see that the modularity objective has a one-step random walk interpretation and a generalized modularity extends this to varying walk lengths [14, 32]. Such a generalized objective in practice is achieved through a resolution hyperparameter in the Louvain algorithm (Algorithm 1) [6], which essentially controls the length of the random walk. Higher resolution values correspond to shorter random walks, resulting in more clusters, smaller cluster sizes, and thus larger embedding tables. Table 4 shows the retrieval task performance of GraphHash under different resolution values, along with a comparison to double frequency hashing (the strongest baseline).

From Table 4 we can observe that GraphHash consistently outperforms double frequency hashing at all resolution levels for the

**Table 4: Impact of resolution in modularity clustering objective on model performance in retrieval. GraphHash consistently outperforms double frequency hashing across all resolution levels considered.**

| resolution | | GraphHash | | | double frequency | | |
|---|---|---|---|---|---|---|---|
| | | # param | Recall (↑) | NDCG (↑) | # param | Recall (↑) | NDCG (↑) |
| MF | 50 | 0.212M | **7.296** | **3.798** | 0.216M | 3.064 | 1.809 |
| | 100 | 0.432M | **8.894** | **4.908** | 0.436M | 3.253 | 2.045 |
| | 200 | 0.742M | **9.393** | **5.448** | 0.744M | 3.927 | 2.544 |
| | 400 | 1.140M | **9.733** | **6.032** | 1.140M | 4.521 | 2.964 |
| LightGCN | 50 | 0.212M | **15.365** | **9.766** | 0.216M | 5.105 | 3.453 |
| | 100 | 0.432M | **15.783** | **9.950** | 0.436M | 7.131 | 4.678 |
| | 200 | 0.742M | **15.388** | **9.661** | 0.744M | 10.069 | 6.509 |
| | 400 | 1.140M | **15.289** | **9.531** | 1.140M | 11.698 | 7.563 |

retrieval task. Moreover, while increasing resolution (and thus embedding table size) improves performance for both GraphHash and the baseline with the MF backbone, the resolution value has little effect when using GraphHash with the LightGCN backbone, unlike double frequency hashing which is more sensitive. A similar set of experiments for the CTR task can be found in Appendix D, where DoubleGraphHash consistently outperforms double frequency hashing across different resolution levels.

*6.4.2 Other types of clustering objective.* We also compare to other types of bipartite graph clustering methods, such as the spectral bipartite graph co-clustering proposed in [15][4]. The results are presented in Table 5 for the retrieval task on Gowalla. We see that while the spectral co-clustering method slightly outperforms GraphHash in retrieval, the cost is at the clustering time, where spectral co-clustering requires >170x more time on Gowalla, making it inefficient, if not non-applicable to large-scale graphs. A similar set of experimental results for the CTR task, can be found in Appendix D, where DoubleGraphHash outperforms its spectral variant where the graph clustering component is replaced with spectral co-clustering, in addition to requiring much less clustering time.

**Table 5: Impact of the type of clustering objective on model performance in retrieval. While spectral co-clustering slightly outperforms GraphHash, it requires >170x more time.**

| | | Gowalla | | | |
|---|---|---|---|---|---|
| | | time | # param | Recall (↑) | NDCG (↑) |
| MF | spectral | 332.062s | 0.545M | **9.622** | **5.762** |
| | GraphHash | **1.928s** | 0.542M | 9.144 | 5.273 |
| LightGCN | spectral | 332.062s | 0.545M | 13.088 | **8.0913** |
| | GraphHash | **1.928s** | 0.542M | **13.105** | 8.0731 |

## 7 Conclusion

In this paper, we introduce GraphHash, a novel embedding table reduction method utilizing modularity-based bipartite graph clustering to generate user/item bucket assignments. GraphHash is an efficient alternative to message-passing by using the graph during preprocessing. Empirical evaluation shows the superior performance of GraphHash and its variant in both retrieval and CTR tasks, as well as the robustness of its design choices under various settings. Building upon the promising results of this new graph-based approach, future work could explore how to incorporate the frequency information with graph clustering to better leverage this crucial information [19, 51], and how to adapt GraphHash to the OOV setting [41].

---

[4]We use the implementation provided in the scikit-learn library.

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

# A  Proof of Proposition 3.1

PROOF.  Note that given a graph, the procedures in the Louvain algorithm [6] iterate through the nodes in the order by their indices and thus outputs deterministic clusters (the randomness in its actual implementation in scikit-network, exactly comes from shuffling the node indices at the beginning). Then by putting the users/item cluster assignments in order indexed by their unique IDs, the re-labelling function $\ell$ guarantees that GraphHash is a deterministic function.  □

*Note.* Empirically, we observe that the cluster assignments given by the Louvain algorithm are quite stable even with different random seeds.

# B  Datasets

In this section, we provide detailed description for datasets used in the experiments.

*Data Splitting.* For the context-free top-k retrieval task, we consider the following three datasets: Gowalla [12], Yelp2018 and AmazonBook [1]. For each dataset, we adopt a random split of 80%/10%/10% for training, validation and testing [27]. For the context-aware CTR task, we consider the following three datasets: Frappe [4], MovieLens-1M and MovieLens-20M [24]. For Frappe, we use the split provided in RecZoo[5], where the data are divided into 70%/20%/10% for training, validation and testing [11]. For MovieLens-1M and MovieLens-20M, we adopt a random split of 80%/10%/10% [44].

*Preprocessing.* To avoid out-of-vocabulary (OOV) IDs, which is outside the scope of this work, we preprocess each dataset to satisfy the transductive setting where all users and items in the validation and test sets appear during training. For MovieLens-20M, we use the movie's genres and the user's top-15 tags as the feature information. To make MovieLens-1M and MovieLens-20M suitable for the CTR task, we follow the procedures in [44] such that all the ratings smaller than 3 are normalized to be 0s, all the ratings greater than 3 to be 1s, and rating 3s are removed.

Table 6 and Table 7 summarize the statistics of each dataset used in the retrieval tasks and CTR tasks, respectively.

**Table 6: Datasets used in the retrieval task**

| Dataset | # Users | # Items | # Interactions |
|---|---|---|---|
| Gowalla | 29,858 | 49,981 | 1,027,370 |
| Yelp2018 | 31,668 | 38,048 | 1,561,406 |
| AmazonBook | 52,643 | 91,599 | 2,984,108 |

**Table 7: Datasets used in the CTR task**

| Dataset | # Users | # Items | # Features | # Interactions |
|---|---|---|---|---|
| Frappe | 954 | 4,082 | 8 | 288,609 |
| MovieLens-1M | 6,040 | 3,641 | 48 | 738,983 |
| MovieLens-20M | 138,484 | 24,689 | 35 | 15,709,070 |

[5]https://github.com/reczoo/Datasets

# C  Experimental Setup

*Hyperparameters.* Following [25], we set the embedding dimension to be 64 for all the datasets except MovieLens-20M, in which case the embedding dimension is 32 instead. For MF, we use full batch size and for all the other backbone methods, we use a batch size of 1024 on all the datasets other than MovieLens-20M, where we set the batch size to be 32768 to speed up training. For each set of experiments, we perform a grid search in the following ranges:

- learning rate: $\{1e^{-2}, 5e^{-3}, 1e^{-3}\}$
- weight decay: $\{1e^{-4}, 1e^{-6}, 1e^{-8}\}$

*Optimizer.* We use the Adam optimizer [29].

*Early Stopping.* We use patience of 50 epochs for the retrieval tasks and 5 epochs for the CTR tasks.

*Compute.* We run each experiment using a single NVIDIA Volta V100 GPU with 32GB RAM.

# D  Additional Experimental Results

## D.1  The Impact of Training Objective

We conduct an additional set of experiments with varying values of $\gamma$ in $[0.25, 0.5, 1, 2, 5]$, the strength of the uniformity term in the DirectAU loss, and compare the model performance without hashing, with double frequency hashing (the strongest baseline) and with GraphHash in terms of NDCG@20 on Gowalla and Yelp2018. The results, presented in Figure 5, show that while both the model without hashing and GraphHash are quite robust to changes in $\gamma$, there exists a specific sweet spot for the value of $\gamma$ under double frequency hashing.

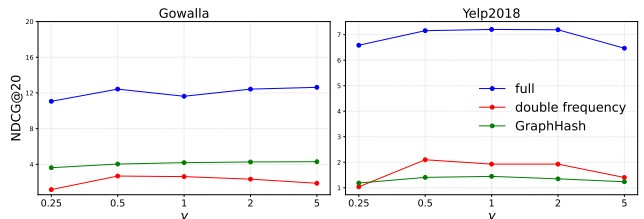

**Figure 5: The impact of the strength of uniformity term $\gamma$ in DirectAU on the model performance. While the model without hashing (full) and GraphHash are quite robust to $\gamma$, there exists a sweet spot in the value of $\gamma$ for double frequency hashing. This suggests that although hashing methods in general might not be as compatible with DirectAU as with BPR (Table 1), GraphHash makes hashing more robust to the choice of $\gamma$.**

## D.2  The Impact of Backbone GNN's Depth

We vary the depth of the LightGCN backbone model and the resulting performance without hashing, with random hashing, and with GraphHash in terms of NDCG@20 on Gowalla and Yelp2018 are shown in Figure 6. The trends are similar to the ones in Figure 4.

## D.3  The Impact of Clustering Objective

*D.3.1  Modularity-based clustering at varying resolution.* The performance of DoubleGraphHash for the CTR task under different

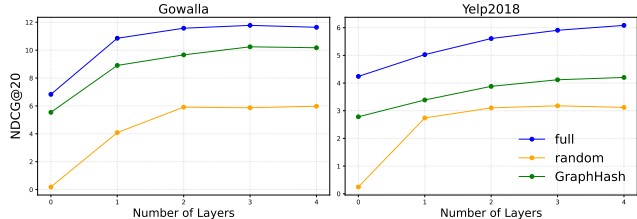

**Figure 6: The impact of LightGCN's depth on the performance of different hashing methods. GraphHash consistently outperforms random hashing. In particular, GraphHash without any additional message-passing layers, performs roughly equal to random hashing with one or two message-passing layers, where the performance in the latter model can be sorely attributed to pure message-passing.**

resolution values are shown in Table 8. For comparison, we also report the performance of double frequency hashing (the strongest baseline), for which the backbone models have roughly similar but strictly larger size compared to the one used for DoubleGraphHash.

**Table 8: Impact of resolution in modality clustering objectives on model performance in CTR task. DoubleGraphHash consistently outperforms double frequency hashing across various resolutions and corresponding embedding table reduction levels.**

| resolution | | DoubleGraphHash | | | double frequency | | |
|---|---|---|---|---|---|---|---|
| | | # param | LogLoss (↓) | AUC (↑) | # param | LogLoss (↓) | AUC (↑) |
| DLRM | 3 | 0.140M | **0.215** | **0.971** | 0.144M | 0.263 | 0.968 |
| | 5 | 0.152M | **0.222** | **0.972** | 0.155M | 0.296 | 0.969 |
| | 10 | 0.171M | **0.259** | **0.975** | 0.174M | 0.306 | 0.972 |
| | 20 | 0.186M | **0.208** | 0.972 | 0.188M | 0.293 | **0.973** |
| DCNv2 | 3 | 0.986M | **0.194** | **0.972** | 0.989M | 0.219 | 0.966 |
| | 5 | 0.998M | **0.194** | **0.972** | 1.001M | 0.208 | 0.970 |
| | 10 | 1.017M | **0.188** | **0.972** | 1.020M | 0.208 | 0.972 |
| | 20 | 1.032M | **0.187** | **0.972** | 1.034M | 0.201 | 0.971 |

In Table 8, we observe that for the CTR task, DoubleGraphHash consistently outperforms double frequency hashing across different resolution levels, similar to the case for the retrieval task reported in Table 4 in the main text.

*D.3.2 Other types of clustering objective.* We compare the results obtained under modularity-based clustering to spectral bipartite graph co-clustering proposed in [15]. The results are presented in Table 9 for the CTR task on Frappe. DoubleGraphHash outperforms its spectral variant, where the clustering component is replaced with spectral co-clustering, while only requiring less than 1/9 of the clustering time of the latter.

**Table 9: Impact of different types of clustering objectives on model performance in CTR task. DoubleGraphHash outperforms its spectral variant, in addition to only requiring <1/9 clustering time.**

| | | Frappe | | | |
|---|---|---|---|---|---|
| | | time | # param | LogLoss (↓) | AUC (↑) |
| DLRM | double spectral | 4.367s | 0.155M | 0.269 | 0.970 |
| | DoubleGraphHash | **0.482s** | 0.152M | **0.222** | **0.972** |
| DCNv2 | double spectral | 4.367s | 1.001M | 0.208 | 0.968 |
| | DoubleGraphHash | **0.482s** | 0.998M | **0.194** | **0.972** |

