# OpenReview forum: "GraphHash: Graph Clustering Enables Parameter Efficiency in Recommender Systems"
_ACM.org/TheWebConf/2025/Conference — WWW 2025 Oral_

### Official Review · Reviewer_wFbp · 2024-11-07

**Novelty:** 7
**Technical Quality:** 7

**Review:**

Summary:

The research introduces GraphHash, a novel approach for reducing the size of embedding tables in recommender systems by leveraging user-item interaction graphs through efficient bipartite graph clustering. Traditional methods, such as the "hashing trick," often lead to collisions that degrade model performance, particularly in tasks like CTR prediction. GraphHash addresses this issue by integrating graph information, which enhances the robustness and adaptability of the model across different user groups and tasks. The study conducts extensive experiments to validate the effectiveness of GraphHash against various baseline hashing methods, demonstrating its superior performance in retrieval tasks and its ability to mitigate the negative impacts of collisions in CTR tasks. Overall, GraphHash represents a significant advancement in the field of recommendation systems by combining graph-based techniques with embedding table reduction strategies.

Pros:

P1. The paper discusses how to utilize hashing techniques to reduce space complexity of graph-based recommendation tasks, which is an interesting topic. A theoretical analysis the connection between the modularity objective and the message-passing mechanism, strengthening the motivation of the proposed method.

P2. Very detailed methodology descriptions with well-justified motivations are provided.

P3. Very extensive experiments are conducted to reveal the superiority of the GraphHash.

P4. The overall content of this paper is well-organized.


Cons:

C1. The authors can introduce more about modularity in the related work instead of general graph clustering.

C2. A overview should be given at the beginning of the methodology part, which can help the readers have a general impression at the first glance so that they can better connect each modules in the following.

**Questions:**

Please respond to the Cons.

**Reviewer Confidence:**

3: The reviewer is confident but not certain that the evaluation is correct

**Scope:**

4: The work is relevant to the Web and to the track, and is of broad interest to the community

---

### Official Review · Reviewer_PtvC · 2024-12-01

**Novelty:** 6
**Technical Quality:** 5

**Review:**

This paper presents a novel hashing-based approach for decreasing the memory requirements of embedding tables in recommender systems. GraphHash leverages modularity-based clustering over the bipartite user-item interaction graph to group similar users and items into buckets. The approach provides a more data-driven alternative to traditional hashing methods by basing the clustering on interaction patterns.

Pros:
- The paper is well-written and structured.
- The use of graph-based clustering for embedding table reduction is both novel and practically relevant in recommender systems.
- Extensive experiments show the effectiveness of GraphHash across various datasets and tasks, including retrieval and click-through rate (CTR) prediction.
- GraphHash is designed to be a plug-and-play solution, requiring minimal changes to integrate into existing recommender systems.

Cons:
- The pseudocode for Algorithm 1 is not explained adequately in the paper.
- There are stronger algorithms and more competitive datasets available for the selected tasks. For top-k item prediction, methods like iALS and RecVAE could have been included as baselines, as they are known to outperform BPR and its variants in many scenarios. Popular datasets like Million Songs and Pinterest are missing. These are frequently used in recommender system evaluations and would provide a more comprehensive comparison. See: Steffen Rendle, Walid Krichene, Li Zhang, and Yehuda Koren. 2022. Revisiting the Performance of iALS on Item Recommendation Benchmarks. In Proceedings of the 16th ACM Conference on Recommender Systems (RecSys '22).
- For the CTR prediction task, the authors should have considered benchmarks from the [CTR papers with code leaderboard](https://paperswithcode.com/task/click-through-rate-prediction) to ensure fair comparisons with state-of-the-art methods.
- The paper provides little discussion about the challenges of deploying GraphHash in real-time systems, where user-item interactions and embedding tables can change dynamically.
- Code for reproduction is not provided.

**Questions:**

- RQ3 requires further motivation. If GraphHash is designed as a plug-and-play solution, why does its performance depend on the choice of the training objective? Can the authors elaborate on scenarios where certain objectives might affect its applicability or effectiveness?
- The proposed method relies on graph clustering as a preprocessing step. How practical is this approach in real-time recommender systems, where user and item graphs are continuously evolving? What strategies could be employed to adapt GraphHash dynamically in such environments?
- Collisions are identified as a critical limitation for CTR tasks. Can the authors explain in more detail why this issue is particularly significant for CTR compared to retrieval tasks? What are the trade-offs between enhancing collision mitigation and maintaining computational efficiency, and how might these trade-offs influence deployment decisions?
- Can GraphHash be extended to incorporate contextual features, such as timestamps, locations, or session data, alongside interaction data? If so, how might these extensions impact the clustering process and the overall performance of the method?

**Reviewer Confidence:**

4: The reviewer is certain that the evaluation is correct and very familiar with the relevant literature

**Scope:**

4: The work is relevant to the Web and to the track, and is of broad interest to the community

---

### Official Review · Reviewer_XaBD · 2024-12-02

**Novelty:** 4
**Technical Quality:** 4

**Review:**

This paper introduces a graph-based approach named GraphHash that leverages modularity-based bipartite graph clustering on user-item interaction graphs to reduce embedding table sizes for industry settings.

pros：

1.	The article presents a clear and coherent narrative, offering a comprehensive and detailed overview of the background and advancements in the field.
2.	The research has practical value.
3.	The experimental analysis covers many interesting aspects, particularly the applicable scenarios of the model, and provides clear explanations.

cons：

1.	The paper does not discuss the current practical applications of hashing methods, including whether the performance degradation caused by hashing is acceptable, or whether there are any real-world applications that have adopted hashing methods.
2.	The paper does not provide information about the datasets used, including their scale, which makes it difficult to demonstrate the applicability of the proposed GraphHash across different dataset sizes.
3.	From Table 2, it can be observed that DoubleGraphHash does not consistently outperform the second runner in the CTR task, which undermines the demonstrated effectiveness of DoubleGraphHash.

**Questions:**

1.	Could you provide a more detailed background, including whether hashing methods are widely accepted in practical scenarios and the performance of the latest hashing methods?

2.	Could you provide more detailed information about the datasets to demonstrate the generalizability of GraphHash?

**Reviewer Confidence:**

3: The reviewer is confident but not certain that the evaluation is correct

**Scope:**

4: The work is relevant to the Web and to the track, and is of broad interest to the community

---

### Official Review · Reviewer_AC7v · 2024-12-02

**Novelty:** 4
**Technical Quality:** 5

**Review:**

With the aim for optimizing embedding table reduction, this paper introduces a new method, named GraphHash, which utilises the graph-based collaborative information. Specifically, GraphHash leverages modularity-based bipartite graph clustering on user-item interaction graphs to reduce embedding table sizes.

The paper is well-written, and easy to follow. While the idea is new, it is not surprising that using the user-item graph information can help to reduce the embedding table, and the question is what is the cost of this? However, this is not discussed in this paper.

pros:
1. The paper is well-written and easy to follow.
2. The paper is addressing an important problem.

cons:
1. the proposed idea seems straightforward.
2. the evaluation seems unfair when comparing with weak baselines.
3. the evaluation is not complete, as it is unclear how the proposed method perform on specific user group (e.g. the inactive users and unpopular items, both of which are likely to be the cost as mentioned above).

**Questions:**

Given the goal of this research is to optimise the embedding table reduction, it is required to compare with other relevant methods, not just those hashing methods, e.g.
* Geet Sethi, Bilge Acun, Niket Agarwal, Christos Kozyrakis, Caroline Trippel, and Carole-Jean Wu. 2022. RecShard: statistical feature-based memory optimization for industry-scale neural recommendation. In Proceedings of the 27th ACM International Conference on Architectural Support for Programming Languages and Operating Systems (ASPLOS '22). Association for Computing Machinery, New York, NY, USA, 344–358. https://doi.org/10.1145/3503222.3507777
* Fuyuan Lyu, Xing Tang, Hong Zhu, Huifeng Guo, Yingxue Zhang, Ruiming Tang, and Xue Liu. 2022. OptEmbed: Learning Optimal Embedding Table for Click-through Rate Prediction. In Proceedings of the 31st ACM International Conference on Information & Knowledge Management (CIKM '22). Association for Computing Machinery, New York, NY, USA, 1399–1409. https://doi.org/10.1145/3511808.3557411
* Xurong Liang, Tong Chen, Lizhen Cui, Yang Wang, Meng Wang, and Hongzhi Yin. 2024. Lightweight Embeddings for Graph Collaborative Filtering. In Proceedings of the 47th International ACM SIGIR Conference on Research and Development in Information Retrieval (SIGIR '24). Association for Computing Machinery, New York, NY, USA, 1296–1306. https://doi.org/10.1145/3626772.3657820
* Yunke Qu, Tong Chen, Quoc Viet Hung Nguyen, and Hongzhi Yin. 2024. Budgeted Embedding Table For Recommender Systems. In Proceedings of the 17th ACM International Conference on Web Search and Data Mining (WSDM '24). Association for Computing Machinery, New York, NY, USA, 557–566. https://doi.org/10.1145/3616855.3635778
* Hailin Zhang, Zirui Liu, Boxuan Chen, Yikai Zhao, Tong Zhao, Tong Yang, and Bin Cui. 2024. CAFE: Towards Compact, Adaptive, and Fast Embedding for Large-scale Recommendation Models. Proc. ACM Manag. Data 2, 1, Article 51 (February 2024), 28 pages. https://doi.org/10.1145/3639306
Furthermore, the comparison with reference [19, 51] seems unfair, as they are mainly proposed to address the collision problem (e.g. in [46]). It is also expected that the proposed GraphHash can produce better results as it utilised extra information (e.g. the user-item graph), and it took more time to process and to learn from the graph. So, a fair comparison shuold consider this extra cost in time as well. So, a comparison in time is needed. Moreover, the clustering process will need to be repeated when there are new user-item interaction data, but there is no such extra re-calculation cost for [19, 51]. Overall, current baselines seem weaker, and stronger ones should be used as mentioned above.

The clustering process potentially impacts the granularity of individual embeddings negatively, and the possible consequence is GraphHash’s performance on unpopular items and inactive users will become worse. This should be discussed and investigated in the experiments.

Not just the clustering objectives, the clustering algorithms (and its corresponding clustering quality) are likely to affect the performance of the proposed method. So, it is needed to explore and measure to what quality of clustering would lead to better performance. How about other clustering algorithms?

GraphHash conducted fully smoothing node embeddings within the same cluster. Is this over simplified? If not, please explain and discuss why, and the corresponding experiment should be carried out as well.

In addition, the quality of the user-item interaction graph has significant impact to the performance of the proposed GraphHash. A consequent question is: when the user-item interaction data/graph is extremely sparse, how will this affect the performance of HashGraph?

Minor:
Line 65: what does ‘3B’ means? 3 Billion?

**Reviewer Confidence:**

4: The reviewer is certain that the evaluation is correct and very familiar with the relevant literature

**Scope:**

4: The work is relevant to the Web and to the track, and is of broad interest to the community

---

### Official Review · Reviewer_btey · 2024-12-03

**Novelty:** 5
**Technical Quality:** 4

**Review:**

Pros:
1. This paper is the first paper that proposes a hash method that uses graph clustering to optimize embedded tables. It is innovative and may solve key practical problems.
2. Comprehensive experiments have proven that although the proposed algorithm is relatively simple to implement, its performance in retrieval tasks has been significantly improved.

Cons:
1. The algorithm in this paper relies on u-i interaction. If it is in a cold start scenario, there may be a problem of performance degradation. The paper lacks discussion of cold start scenarios.
2. The main problem solved by this paper is the problem in ultra-large-scale data scenarios, but the data set used in the experiment is limited in size, and no industrial-grade large-scale data set is used to verify the performance of the algorithm.
3. In the CTR task, GraphHash performs worse than DoubleGraphHash, and the method may not be optimized enough for conflict-sensitive tasks (such as AUC).
4. In the CTR task, comparisons with other more complex deep learning CTR models (such as DeepFM, DIN, etc.) can be included.

**Questions:**

1. Have you considered the efficiency of the algorithm in the cold start scenario? Can you provide further arguments in the article?
2. The paper only considers the interaction between users and items. Can other characteristics of users and items be combined to further improve the algorithm?
3. Does DoubleGraphHash have similar performance optimization in the retrieval task as in the CTR task?
4. Has the algorithm been experimented with ultra-large-scale datasets? How is its performance?

**Reviewer Confidence:**

3: The reviewer is confident but not certain that the evaluation is correct

**Scope:**

4: The work is relevant to the Web and to the track, and is of broad interest to the community